# Effective Clinical Pathway Improves Interprofessional Collaboration and Reduces Antibiotics Prophylaxis Use in Orthopedic Surgery in Hospitals in Indonesia

**DOI:** 10.3390/antibiotics11030399

**Published:** 2022-03-16

**Authors:** Fauna Herawati, Adinda Dessi Irawati, Ella Viani, Nully Andaretha Sugianto, Nur Laili Rahmatin, Made Prita Artika, Sukmawati Eka Bima Sahputri, Kevin Kantono, Rika Yulia, Retnosari Andrajati, Diantha Soemantri

**Affiliations:** 1Department of Clinical and Community Pharmacy, Faculty of Pharmacy, Universitas Surabaya, Surabaya 60293, Indonesia; adindadessi30@gmail.com (A.D.I.); ellavian1010@gmail.com (E.V.); nullyandaretha@gmail.com (N.A.S.); laily.rahmatin@gmail.com (N.L.R.); pritaartika100@gmail.com (M.P.A.); sukmawati.sahputri@gmail.com (S.E.B.S.); rika_y@staff.ubaya.ac.id (R.Y.); 2Department of Pharmacology and Clinical Pharmacy, Faculty of Pharmacy, Universitas Indonesia, Depok 16424, Indonesia; retnosaria@gmail.com; 3Laboratory for Developmental Psychology, Faculty of Psychology, Universitas Surabaya, Surabaya 60293, Indonesia; setiasih@staff.ubaya.ac.id; 4Department of Food Science, Auckland University of Technology, Private Bag 92006, Auckland 1142, New Zealand; kevin.kantono@aut.ac.nz; 5Department of Medical Education, Faculty of Medicine, Universitas Indonesia, Depok 16424, Indonesia; diantha.soemantri@ui.ac.id

**Keywords:** interprofessional collaborative practice, antibiotic stewardship, defined daily dose, clinical pathway, antibiotics prophylaxis

## Abstract

Clinical pathways can improve the quality of health services. The effectiveness and impact of implementing clinical pathways are controversial. The preparation of clinical pathways not only enacts therapeutic guidelines but requires mutual agreement in accordance with the roles, duties, and contributions of each profession in the team. This study aimed to investigate the perception of interprofessional collaboration practices and the impact of clinical pathway implementation on collaborative and Defined Daily Dose (DDD) prophylactic antibiotics per 100 bed-days in orthopedic surgery. The Collaborative Practice Assessment Tool (CPAT) questionnaire was used as a tool to measure healthcare’ perceptions of collaborative practice. The clinical pathway (CP) in this study was adapted from existing CPs published by the Indonesian Orthopaedic Association (Perhimpunan Dokter Spesialis Orthopaedi dan Traumatologi Indonesia, PABOI) and was commended by local domestic surgeons and orthopedic bodies. We then compared post-implementation results with pre-implementation clinical pathway data using ANCOVA to explore our categorical data and its influence towards CPAT response. ANOVA was then employed for aggregated DDD per 100 bed-days to compare pre and post intervention. The results showed that the relationships among members were associated with the working length. Six to ten years of working had a significantly better relationship among members than those who have worked one to five years. Interestingly, pharmacists’ leadership score was significantly lower than other professions. The clinical pathway implementation reduced barriers in team collaboration, improved team coordination and organization, and reduced third-generation cephalosporin use for prophylaxis in surgery (pre: 59 DDD per 100 bed-days; post: 28 DDD per 100 bed-days). This shows that the clinical pathway could benefit antibiotic stewardship in improving antibiotic prescription, therefore reducing the incidence of resistant bacteria.

## 1. Introduction

Antibiotic prophylaxis is an antibiotic delivered 60 min (two hours if administered vancomycin) before surgery to a maximum of 24 h after surgery. The benefit of prolonged surgical antibiotic prophylaxis is debatable, where the overuse of antibiotics leads to antimicrobial resistance. This may lead to increased costs of hospitalization, especially if an individual is infected by methicillin-resistant *Staphylococcus aureus* (MRSA). Some studies showed that the rates of infection in patients who received prophylaxis for one day and those who received it for three days or seven days were similar [1]. Nevertheless, a study reported that the number of periprosthetic joint infections in patients with total hip or knee arthroplasties (THA/TKA) without oral surgical prophylaxis was 4–5 times more likely than if they were administered extended oral SAP (surgical antibiotic prophylaxis) [2].

An interprofessional collaboration practice is a multidisciplinary collaboration that integrates nursing care, nutritional care, and pharmaceutical care to improve service quality. Hospital Accreditation Standards describes Clinical Pathways (CP) as a valuable tool in Integrated Care practice to control quality and cost [3,4]. An active collaboration during the implementation of clinical care pathways is a standard of input, process, and outcome that eliminates unnecessary or inefficient treatments [5]. Clinical pathway implementation and hospital formularies also include antimicrobial stewardship (AMS) strategies [6,7]. Research on the benefits of implementing CP on controlling antibiotic use in developing countries has not been widely carried out. The benefit (length of stay, healthcare cost, and service quality) of implementing CP in other patient care (emergency, surgery, and other clinical cases) has controversial results [5,8,9]. A review showed that the average length of stay for hip fractures was reduced, but on the contrary, esophagectomy patients showed no significant difference [5]. A study reported that a successful clinical pathway implementation was related to the cognitive level of hospital administrators and clinical staff, the system of hospital accreditation, and feedback of CP implementation [10].

In a clinical pathway, multidisciplinary decision making increases engagement between professionals for a better quality of care. Shared decision making is often described as the process of making decisions towards patient care based on mutual agreement on clinical evidence and available information [11,12]; it starts with assessing signs and symptoms, determining diagnosis, deciding the severity, and the implementation of therapy [13]. An interprofessional collaboration practice is built by members by taking responsibility for their contributions to the team, interaction, or discussion in providing feedback among team members; communication among team members [13] to achieve better goals; and commitment among team members to success. The treatment decisions are taken by considering all team members’ knowledge and contribution to patient care [14,15].

Research in Japan that assessed the practice of interprofessional collaboration of medical personnel in three hospitals has shown that age (i.e., younger professionals) and profession (e.g., nursing) were the most influential positive factors in creating a collaborative environment [16]. Healthcare practitioners from countries with complementary models (e.g., United States and Israel) possess more positive attitudes to interprofessional collaboration than healthcare practitioners from countries with hierarchical models (e.g., Italy, Mexico, and including Indonesia) [13,17,18,19]. Thus, promoting complementary collaborative models may be necessary to improve attitudes towards collaborative practice. In the complementary model, all professions share responsibilities and have complementary roles concerning patient care. A study in a West Java regional hospital that uses Collaborative Practice Assessment Tool (CPAT) as a tool showed that leadership and decision making were the two dominant factors that influence interprofessional collaboration practice [20]. However, interprofessional collaboration practices in hospitals sometimes are not mandatory and are not supported by management, especially in Indonesia. Interprofessional collaboration practices are strongly needed to improve the quality of healthcare services and patient safety [21].

Along with time, the number of broad-spectrum antibiotic use remains high especially with a hierarchical culture in Indonesia as a barrier to interprofessional collaboration. Hence, the first aim in our study was to explore the perception of interprofessional collaboration practice and how it is influenced by external (e.g., work experience) or internal factors (e.g., age and gender). The second aim was then to implement a clinical pathway to further improve collaboration practice and antibiotic use. Specifically, we aimed to determine the differences of interprofessional collaboration practices perception in orthopedic surgery healthcare at Husada Utama Hospital. This was performed before (i.e., baseline) and after clinical pathway implementation. The impact of CP implementation on the use of antibiotics in hospitals was then assessed. Our hypothesis is that clinical pathways would foster a collaborative environment between health professionals, which results in the judicious use of antibiotics and reduces the incidence of antimicrobial resistance.

## 2. Methods

The first study is aimed to measure the interprofessional collaboration practice perception and whether various covariates such as demographics may contribute to the difference between collaborative perception, whereas the second study then measured the impact of clinical pathways in antibiotic use in a hospital. Two hundred and twenty respondents from three referral hospitals participated in Study I. The Husada Utama Hospital is a private hospital in Surabaya and has 235 beds; Bangil Regional Public Hospital is a public hospital in Pasuruan and has 366 beds; and Hajj General Hospital is a public hospital and has 293 beds. The intervention, clinical pathway, was used for orthopedic healthcare practitioners in Husada Utama Hospital, Surabaya, admitted in December 2020. The Collaborative Practice Assessment Tool (CPAT) instrument was developed to assess the degree of collaboration and identifies the strengths and weaknesses of collaborative practice which then provide opportunities to focus on training interventions for team members [22,23].

### 2.1. Study I

#### Perception of Interprofessional Collaborative Practice

The assessment of healthcare practitioners’ perceptions of collaborative practices was measured using the Collaborative Practice Assessment Tool (CPAT) questionnaire that has been validated in the Indonesian context [13]: the Indonesian version of CPAT. The questionnaire was validated using exploratory factor analysis (EFA) after language adaptation and trial. EFA showed the adequacy of the sample with Measure of Sampling Adequacy (MSA) 0.728–0.965, the Kaiser–Meyer–Olkin (KMO) 0.923, and Bartlett’s Sphericity Test 0.000. The correlation coefficient for 53 questions is >0.3 with a significance level of 5%. The reliability of the CPAT questionnaire was measured with Cronbach’s alpha of 0.977, which consists of eight components with a total of 53 questions (Appendix A): i. relationships among team members (9 questions); ii. barriers to team collaboration (5 questions); iii. team relationships within the community (4 questions); iv. team coordination and organization (14 questions); v. decision making and conflict management (2 questions); vi. leadership (5 questions); vii. missions, goals, and objectives (9 questions); and viii. patient involvement, responsibility, and autonomy (5 questions). The CPAT form (hardcopy) was distributed to nurses and pharmacists by the Husada Utama Hospital, Bangil Regional Public Hospital, and Hajj General Hospital Training and Development Division. Unfortunately, the researchers were not able to meet in person due to COVID-19 pandemic. Each respondent had an invitation by phone and signed a consent form indicating their willingness to participate in this research. Interestingly, data collection with a hardcopy version during the training session for the pre-intervention stage (3 days) was faster than the post-intervention stage (7 days)—this was probably due to internal communication by the head of the department. Three doctors filled an e-form of the questionnaire immediately after receiving a Google form link (Alphabet Inc., Mountain View, CA, USA). One doctor filled a hardcopy questionnaire that was delivered face to face. The questionnaire data collection was carried out to provide CPAT score in three hospitals. 

### 2.2. Study II

#### Research Design

This research is a pretest–posttest one-group design. The intervention in this research was the clinical pathway (CP). This research was conducted from November 2020 to January 2021 in Husada Utama Hospital. The respondents in this study were orthopedic specialists, pharmacists, and nurses who were directly involved in orthopedic patient care. Questionnaire data collection was carried out twice in early December 2020 (pre-test, before CP implementation) and early January 2021 (post-test, after CP implementation).

The intervention used in this study was closed fracture clinical pathway. There were twelve CPs applied. The diagnosis of the CP were closed fracture antebrachii, fracture wrist and hand, fracture of carpal bone, contracture of joint, carpal tunnel syndrome, adhesive capsulitis of shoulder, closed fracture of radius and ulna, closed tibia fracture, osteomyelitis, rupture tendon, soft tissue injury of knee, proximal tibia fracture. The Husada Utama Hospital management had not established Clinical Pathway for orthopedic surgery. The clinical pathway in this study was adapted from existing CP published by the Indonesian Orthopaedic Association (Perhimpunan Dokter Spesialis Orthopaedi dan Traumatologi Indonesia, PABOI) and was commended by local domestic surgeons and orthopedic bodies. CP (in the form of a tick and patient-oriented short note of nursing care, medical actions, nutritional care, etc.) was used as the patient care guide for each orthopedic surgery patient admitted in December 2020. When the patient was finally discharged from the hospital, each existing CP was then signed by the responsible doctor. This documented CP can be then reviewed at any time.

### 2.3. DDD per 100 Bed-Days

Defined Daily Dose (DDD) is the assumed average maintenance dose per day for a drug used for its main indication in adults, a statistical measure of drug consumption [24,25]. The overuse of antibiotics will shift the competitive balance of susceptible and resistant microorganisms (selection pressure); therefore, monitoring and controlling antibiotic use is important. Define daily dose (DDD) is a unit for measuring antibiotic use that is widely use and can be compared internationally. A quantitative evaluation used the DDD per 100 bed-days, which is calculated using the formula below [26]:$$\frac{\mathrm{DDD}}{100}\mathrm{bed}\;\mathrm{days}=\frac{\mathrm{Total}\;\mathrm{Antibiotics}\;\left(\mathrm{gram}\right)\times 100}{\mathrm{DDD}\;\mathrm{WHO}\;\left(\mathrm{gram}\right)\times \mathrm{LOS}}$$
where DDD WHO is the Defined Daily Dose determined by WHO and LOS is the Total Length of Stay. 

### 2.4. Statistical Analysis

In Study I, the analysis of Covariance (ANCOVA) was used to investigate the influence of age, gender, work length, profession, and previous experience in collaborative practice on CPAT responses. Fisher’s Least Significant Difference (LSD) was then applied if significance was observed. In a similar manner, ANCOVA was employed to the dataset that was collected in Study II with the focus on pre-post changes in CPAT perception. Fisher’s Least Significant Difference (LSD) was then applied if significance was observed. In addition, a generalized Analysis of Variance (ANOVA) model was also carried out on the aggregated DDD dataset to identify the changes of DDD and DDD/100 bed-days between pre- and post-CP implementation. All analysis was performed using XLSTAT 2022.1.1 (Addinsoft, New York, NY, USA). 

## 3. Results

From three hospitals that participated in this study, there were 261 healthcare respondents (Section 2.1, Table 1, Table 2 and Table 3): 98 respondents from Husada Utama Hospital (HUH), 96 respondents from Bangil Regional Public Hospital (BRPH), and 67 respondents from Hajj General Hospital (HGH). Table 2 and Table 3 include the influence of the length of employment experience and profession on interprofessional collaboration practice perception. One of the three hospitals, Husada Utama Hospital with 52 participants in the orthopedic department, had agreed to implement clinical pathways and measure its impact on antibiotic use (Section 2.2, Table 4, Table 5 and Table 6). Table 6 represents all antibiotic use in the study period.

### 3.1. Perception of Interprofessional Collaborative Practice

A cross-sectional survey showed that, among respondents, there are 35% aged 30–40, 81% female, 76% nurses and midwives, 34% with work lengths of 6–10 years, and only 4% did not experience in collaborative practice. 

#### Influence of Demographics and Employment Experience

Work length influences relationships among members. Work length was shown to be a significant factor for relationships among members (F_(10,219)_ = 5.521; *p* < 0.01) (Table 2). The highest score was shown for participants that worked for 6–10 years followed by participants that had worked for more than 10 years. A significantly lower score was reported for participants who had worked less than 5 years compared to participants that had 6–10 years experience.

Profession influences CPAT perception. Profession was found to be a significant factor for barriers in team collaboration (F_(10,219)_ = 10.395; *p* < 0.001), leadership (F_(10,219)_ = 9.307; *p* < 0.001), and patient involvement, responsibility, and autonomy (F_(10,219)_ = 17.328; *p* < 0.001) (Table 3). It was shown that nurses are significantly rated the lowest in barriers in team collaboration compared to other professions. A highest score of leadership was reported in doctors, midwides, and nurses. It was significantly lower in pharmacists followed by technicians. The significant lowest score for patient involvement, responsibility, and autonomy was reported by technicians compared to other profession groups. There was no reported effect of age, gender, and previous experience in collaborative practice towards collaborative practice perception.

### 3.2. Influence of Effective Clinical Pathway

The characteristics of the respondents as follows: 60% respondents were 26–35 years, 81% female, 79% nurses, 44% had been working for more than ten years, and 88% experienced collaborating with other professions (Table 4). The fifty-two health care practitioners were four orthopedic specialists, forty one nurses, and seven pharmacists.

CP significantly increases CPAT perception. The score of collaborative practice perceptions in post-intervention (using clinical pathways) was significantly higher for barriers in team collaboration (Cohen’s d = 0.351; medium effect size) and team coordination and organization (Cohen’s d = 0.104; small effect size) compared to baseline (Table 5).

Significant decrease were reported after CP intervention for both DDD (F_(8,15)_ = 9.051; *p* < 0.05) and DDD/100 bed-days (F_(8,15)_ = 9.589; *p* < 0.05) (Table 6). This implies that CP implementation was indeed successful in decreasing DDD and promotes the judicial use of antibiotics in this study.

## 4. General Discussion

### 4.1. Work Length Influences Relationships among Members

There are 34% respondents that have a work length 6–10 years (Table 1 and Table 2). Our results resonate with other studies that had reported work experience affect work ability [27], where the period of employment is significantly associated with good working relationships and knowledge integration problems [28]. A multicenter longitudinal study in 13 hospitals in Germany found that work experience and period of employment was associated with interprofessional collaboration perception (good working relationship), particularly between 1 and 3 months versus 1–5 years, but not more than 5 years. Moreover, perceptions of inter-professional teamwork within wards seemed similar across professional groups due to the impact of ward affiliation. This study suggests training entire inter-professional teams in future interventions [28].

### 4.2. Profession Influences CPAT Perception

In this study, nurses significantly rated the lowest in barriers in team collaboration compared to other professions (Table 3). Nurses often have a close relationship with the patient and play a role in preventing disease complications and are often the first who detect health emergencies, including adverse drug reactions. Nurses’ contributions to medical care and pharmaceutical care will reduce the barrier of nurses’ collaboration with other professions. A systematic review had reported that, in more than 30% (15 out of 50 studies) of the included studies, nurses were heavily involved in interventions for improving patients’ care, especially patients’ adherence towards medication [29]. Pharmacists were reported to have higher barriers in collaboration than nurses because of limited doctor and nurse knowledge about the pharmacists’ competencies. Pharmacists possess clinical skills [30,31] and not only skills in drug management and procurement [32]. Another barrier to collaboration that has been reported between physicians and pharmacists is communication, lack of specific collaboration rules (standards of cooperation), self-confidence, low mutual respect, and trust [31]. Generally, doctors are often recognized as the leaders of clinical teams. Leadership skills are commonly embedded and developed in their medical education and training, both in undergraduate and postgraduate level to take responsibility for the delivery of excellent patient care [33,34]. A leader promotes collaboration across members of the healthcare team, manages resources and maintains staff commitment to getting work performed [35]. However, it has been reported that pharmacists’ expertise remained untapped in the context of interprofessional care, for example, in assisting in the reduction in medical costs for prescription medications and to increase the rationality of therapy for patients [32]. Pharmacy technicians are a part of the pharmacy team. Pharmacists provide clinical services patient care, whereas pharmacy technicians’ tasks are mainly stock management, dispensing, prescription administration (collection and filing and repeat supply) and assisting with audits [36]. In the future, in addition to technical tasks, newly proposed roles include clinical tasks (handing out medicines) and management/training tasks (responding to queries and dealing with complaints) [36,37,38]. The nature of their task in the integrative healthcare system can contribute to a sense of detachment towards patients compared to other professions [39,40,41].

### 4.3. CP Significantly Increases CPAT Perception

In this study, CP intervention showed a small to medium effect on the behavior of healthcare practitioners (Table 5). Clinical pathway implementation is appropriate or effective for most surgeries or high-volume procedures. It is used a tool to ensure effective integration and coordination of services by efficiently using existing resources and is a valued document of Good Clinical Governance in hospitals, which resulted in positive outcomes for patients. However, it is to note that CP requires a multidisciplinary approach or interprofessional collaboration in the integrative healthcare system [42,43]. To provide high-quality care, an institution develops, implements, and evaluates clinical pathways (CPs). Clinical Pathways in the care of patients with a specific clinical problem may reduce variations in clinical practice, perform evidence-based practice [44], and optimize resource allocation and cost-effectiveness [45]. Resonating with our findings, team collaboration was often a reported barrier to a successful CP implementation. Determining the role and commitment of all relevant parties is a key factor in CP implementation success [45,46]. CP implementation is a leader-driven initiative; therefore, the awareness and support of hospital leaders to develop strategic policies can act as a tool in change management, as an integral component in business management and service quality assurance. Hence, awareness, commitment, and the role of senior managers/staff are crucial factors in the successful implementation of CP and to uphold good clinical governance [45].

A decree or support from the director/hospital senior management to support and implement the clinical pathway is important for organizational commitment. Leaders or directors can be an inspiration in work and determine the direction and goals of the organization. Senior leaders can demonstrate their capacity to carefully delegate responsibilities and instill a strong sense of belonging to the organization in their employees. This attitude may influence employees to be able to commit to their organization. The effectiveness of an organization is often determined by the role of leaders who are willing to bring organizational members towards achieving vision, mission, and goals. The leader can provide social effects with a personal approach [47,48], authentic style [49,50], and building of two-way communication [14,51,52,53,54].

The quality of CP that was developed varied. A good CP is translated from an evidence-based best practice, evaluates processes and outcomes regularly, and the awareness of its benefits in other fields [55]. IT-system support is also crucial in implementing an elegant and sophisticated CP [56]. The positive impacts of CP towards interprofessional collaboration that have been reported were (i) professional contribution with respect to their unique competence, roles, task distribution, and responsibilities to complement each other [57,58,59]; (ii) reduction in the amount of time for communication, shared information, planning, and decision making [57,58]; (iii) the dependence and recognition between profession in the integrative healthcare systems [60]; and, finally, (iv) to foster organization and collaboration culture model [60].

### 4.4. DDD Decreases after CP Intervention

This study reported a (near) fifty percent DDD per 100 bed-days reduction (Table 6). Similarly to other studies, clinical pathways increase the clinical appropriateness of antibiotics [61,62,63] and reduces broad-spectrum antibiotics regimen, accompanied with fewer antibiotic courses [64].

### 4.5. Limitations and Future Research

This study was unfortunately carried out during the height of COVID-19 pandemic, and the researchers were not able to interact with healthcare professionals in the hospital due to physical distancing rule. We, therefore, were only able to provide intervention in one hospital. Other limitations in this study were that the provision of intervention to health workers was not directly to every healthcare practitioner but by seminars or CP material debriefs due to workplace protocols to mitigate and control the transmission of COVID-19. The limited social engagement aspect of this study’s CP implementation towards healthcare practitioner may result in low adherence towards CP compliance, where adherence and compliance have been reported to be strongly associated [65,66]. Knowledge from CP intervention is often not always followed by behavioral changes, especially when pressure from external factors (egalitarianism, facilities, or systems) such as forcing the subject to change behavior [67,68].

## 5. Conclusions

A clinical pathway is a standard operating procedure (SOP) that combines orthopedic specialists, pharmacists, and nurses’ care for the patients built by mutual agreement of each profession in the team in terms of roles, duties, and contributions. It is an evidence-based protocol that complies with therapeutic guidelines that includes essential multidisciplinary care steps in inpatient care. Our study showed that collaboration practices were significantly influenced by work length and profession. The implementation of clinical pathway showed significant improvement in interprofessional collaboration practices, particularly in perceived barriers and team coordination. The positive improvement of such practices also resulted in a reported decrease in DDD profile in orthopedic patients. Our study showed the benefit and calls for the implementation of clinical pathway in Indonesian hospitals.

## Figures and Tables

**Table 1 antibiotics-11-00399-t001:** CPAT respondent demography characteristic for Study I (N = 261 − N_HUH_ = 98; N_BRPH_ = 96, N_HGH_ = 67).

Characteristics	Frequency	Percentage (%)
Age		
21–25 years	25	11.36
26–30 years	62	28.18
31–35 years	51	23.18
36–40 years	27	12.27
41–45 years	17	7.73
>45 years	38	17.27
Gender		
Male	42	19.09
Female	178	80.91
Profession		
Doctor	29	13.18
Pharmacist	14	6.36
Nurse	87	39.55
Midwife	80	36.36
Technician	10	4.55
Work length		
1–5 years	71	32.27
6–10 years	74	33.64
>10 years	75	34.09
Experience in collaborative practice		
Yes	212	96.36
No	8	3.64

**Table 2 antibiotics-11-00399-t002:** Reported CPAT scores categorized by work length (N = 261 − N_HUH_ = 98; N_BRPH_ = 96, N_HGH_ = 67).

Work Length	*p*-Value	1–5 Years	6–10 Years	>10 Years
Relationships among members	<0.001	4.228 b	4.472 a	4.417 ab
Barriers in team collaboration	0.210	3.368 a	3.615 a	3.506 a
Team relationships with the community	0.252	3.580 a	3.714 a	3.853 a
Team coordination and organization	0.698	4.253 a	4.306 a	4.269 a
Decision making and conflict management	0.684	1.835 a	1.771 a	1.712 a
Leadership	0.627	3.808 a	3.883 a	3.830 a
Mission, goals, and objectives	0.294	4.008 a	4.125 a	4.080 a
Patient involvement, responsibility, and autonomy	0.353	3.680 a	3.829 a	3.811 a

a,b means with different letters show the significant effect of work length based on Fisher’s Least Significant Difference (LSD), posthoc grouping based on multiple comparisons.

**Table 3 antibiotics-11-00399-t003:** Reported CPAT scores categorized by profession (N = 261 − N_HUH_ = 98; N_BRPH_ = 96, N_HGH_ = 67).

Profession	*p*-Value	Doctor	Midwife	Nurse	Technician	Pharmacist
Relationships among members	0.353	4.435 a	4.470 a	4.357 a	4.355 a	4.244 a
Barriers in team collaboration	<0.001	3.661 a	3.808 a	2.865 b	3.560 a	3.589 a
Team relationships with the community	0.053	3.911 a	3.869 a	3.884 a	3.291 a	3.624 a
Team coordination and organization	0.786	4.366 a	4.275 a	4.280 a	4.285 a	4.172 a
Decision making and conflict management	0.414	1.855 a	1.913 a	1.775 a	1.661 a	1.659 a
Leadership	<0.001	4.144 a	4.072 a	4.085 a	3.152 c	3.750 b
Mission, goals, and objectives	0.216	4.222 a	4.147 a	4.139 a	3.929 a	3.917 a
Patient involvement, responsibility, and autonomy	<0.001	4.110 a	4.068 a	4.146 a	2.571 b	3.971 a

a,b,c means with different letters show the significant effect of work length based on Fishers Least Significant Difference (LSD), posthoc grouping based on multiple comparisons.

**Table 4 antibiotics-11-00399-t004:** CPAT respondent demography characteristic for Study II (N_HUH_ = 52).

Characteristics	Frequency	Percentage (%)
Age		
21–26 years	5	9.62
26–31 years	12	23.08
31–35 years	19	36.54
>35 years	16	30.77
Gender		
Male	10	19.23
Female	42	80.77
Profession		
Doctor specialist	4	7.69
Pharmacist	7	13.46
Nurse	41	78.85
Work length		
1–5 years	9	17.31
5–10 years	20	38.46
>10 years	23	44.23
Experience in collaborative practice		
Yes	46	88.46
No	6	11.54

**Table 5 antibiotics-11-00399-t005:** Overall perception of interprofessional collaboration practices before and after clinical pathway intervention (N_HUH_ = 52).

Condition	*p*-Value	Pre	Post	Effect Size (Cohen’s d)
Relationships among members	0.229	4.278 a	4.252 a	-
Barriers in team collaboration	<0.001	3.112 b	3.442 a	0.351
Team relationships with the community	0.390	3.837 a	3.904 a	-
Team coordination and organization	<0.05	4.082 a	4.016 b	0.104
Decision making and conflict management	0.159	1.885 a	1.923 a	-
Leadership	0.322	4.231 a	4.238 a	-
Mission, goals, and objectives	0.991	4.211 a	4.211 a	-
Patient involvement, responsibility, and autonomy	0.159	4.115 a	4.269 a	-

a,b means with different letters show the significant effect of work length based on Fisher’s Least Significant Difference (LSD), posthoc grouping based on multiple comparisons.

**Table 6 antibiotics-11-00399-t006:** Profile of DDD 100 bed-days orthopedic patients.

NO	ATC Code	Antibiotic Name	Pre		Post	
			DDD	DDD/100 Bed-Days	DDD	DDD/100 Bed-Days
ORAL						
1	J01DB05	Cefadroxil	533.25	32.24	75.50	17.81
2	J01DD08	Cefixime	446.00	26.96	87.25	20.58
3	J01MA02	Ciprofloxacin	101.00	6.11	-	-
4	J01MA12	Levofloxacin	9.00	0.54	-	-
PARENTERAL						
1	J01DD04	Ceftriaxone	526.00	31.80	26.25	6.19
2	J01DB04	Cefazolin	626.67	37.89	121.50	28.66
3	JO1GB03	Gentamicin	36.27	2.19	-	-
4	J01DD01	Cefotaxime	-	-	4.00	0.94
	Total oral and parenteral		2278.18	137.73	314.5	74.18
	Period		Sept.–Nov. 2020		Jan. 2021	
	Number of patients		337		41	
	Length of stays (days)		1654		424	

## Data Availability

The authors confirm that the data supporting the findings of this study are available on request.

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
