# Peer review of "Effective Clinical Pathway Improves Interprofessional Collaboration and Reduces Antibiotics Prophylaxis Use in Orthopedic Surgery in Hospitals in Indonesia"

_antibiotics, 2022, doi:10.3390/antibiotics11030399_

Round 1

Reviewer 1 Report

The present manuscript described the efficacy of integrated clinical pathway intervention implemented in hospital setting. In particular, this study showed that there was a tendency to decrease the use of prophylactic antibiotics. However, the preparation of integrated clinical pathways requires mutual agreement in accordance with the roles, duties, and contributions of each profession in the team. This is a very important issue to improve the quality of patient treatment in hospitals.

Authors conclude that an integrated clinical pathway is an evidence-based protocol, include details of essential multidisciplinary care steps in inpatient care. However, as reported by authors the present study has some limitations widely described at the end of the text. In light of this, my suggestion is to change the title of manuscript and better describe the conclusions of the work. Authors need to clarify the preliminary aspect of the protocol applied highlighting that it needs to be extensively implemented also in other structures. In other words, authors need to present this work as a proof of concept of integrated clinical pathway intervention implementation. After this minor revisions, the manuscript can be accepted for the publication.

Author Response

We'd like to thank the reviewers for their comments. Please see attached for response and comments.

Reviewer 2 Report

The authors of the paper aimed to investigate the impact of the integrated clinical pathway (ICP) on the perception of interprofessional collaboration practices and their impact on the collaborative 
and the DDD prophylactic antibiotics per 100 bed-days  at  pre-and post-implementation of (ICP) 

Clinical pathways and integrated Clinical pathways  are used interchangably

The title "Effective clinical pathway improves interprofessional collaboration and reduces antibiotics prophylaxis uses in hospitals."

Indicate the intention to reduce antibiotic prophylaxis through the use of ICP intervention,

However, much could be done to make this paper give the intended direction to the reader.

Abstract
In the abstract line 22, the word DDD needs to be defined
Line 24-26:  it is not clear what the authors mean by "the ICP was made by adopting the integrated clinical pathway of other hospitals"
Which are these other hospitals, and who was targeted by the ICP and how?

Line 27: it is unclear which groups the non-statistically significant results refer to!
The abstract missed to report the main findings and the conclusion the does not relate to the results reported 

If you read the abstract and title alone, you do not get the study's location; this is important because the findings from 
one country or a few hospitals can not be generalized globally.

Introduction
Generally, the introduction is biased towards collaboration practice and then a bit on integrated clinical pathways.
It largely ignores the issue of antibiotics prophylaxis, which was the practice for which an ICP intervention is based.

Line 37: choose one word for practice/care
Line 38: A reference is required at the start of the line
Line 39: a reference is necessary to back up the fact
Line 42 Check spelling and grammar of the sentence

Line 44-45 contradicting statements about lack of studies on ICP on antibiotic use and at the same claiming controversy among studies
Line 57 Insert reference for statements you make
Line 82 By America, do you mean the United States? If so, state as such
Line 84 Reference is missing
Line 93-95 states the aim to study perception in orthopedic surgery

The paragraph contradicts the title, which looks at antibiotic prophylaxis, while here orthopedic surgery is mentioned instead

Also, only one hospital is mentioned. The country or region where the study was done is not mentioned.

The link of antibiotic prophylaxis is missing in the background and seems to have ectopically been introduced in the title.
And thus, the introduction fails to highlight the focus of the study clearly.

Methods

The need for having two separate studies in this paper is not obvious
There is no link to the two studies; therefore, the focus of the article is not clear

The study area and setting description is missing
The study design is inconsistently stated
Line 105- 126
Collaborative-Practice Assessment Tool (CPAT) questionnaire is said to have been used to Assess
healthcare practitioners' perceptions of collaborative practices

However, it is unclear when and how the intervention was introduced before this assessment!

Line 137-152: The manner the CP intervention is administered is not fluent and clear to the reader.

Line 153-158
The meaning of DDD is missing
The formula has no any reference given
It is not clear why the calculation is done

Line 159: The statistical method must be clearly stated

Results
It is not clear which hospital the results table 1-3 belong

From Which hospital are the results in Table 4-6?
Table 2

Provide the exact p-value where stated as NS

State clearly what are the figures in Tables 2 and 3

Table 4 is repetition.

Check table 4 gender percentage has a typo.

Table 5 is repetition.

Table 6 why only a few antibiotics were assessed

Why is the intervention period so short

Discussion

Line 225-325
In most of the discussion, The authors discuss the results of other studies without reflecting their own results.
It is unclear when the authors refer to their own findings and when discussing other findings from the literature.

The conclusion is not based on authors study findings

The appendix could be attached as supplementary material instead of being part of the article.

Author Response

(The authors gave the same response as above.)

Round 2

Reviewer 2 Report

The authors provided a revised manuscript of the paper aimed to investigate the impact of the integrated clinical pathway (ICP) on the perception of interprofessional collaboration practices and their impact on the collaborative and the DDD prophylactic antibiotics per 100 bed-days  at  pre-and post-implementation of (ICP) 

There is an improvement compared to the original submission
However, they can improve the caption of tables, the discussion and the conclusion before consideration for publication as follows

Results
The table caption must bring clarity to which hospital the results table 1-3 belong [add tables and total numbers N]

Do the same for Tables 4-6?

Discussion

The discussion can be improved further especially when stating own results and comparing with others.

The conclusion is not based on the author's study findings, improve with the implications of significant findings from your study

The appendix could be attached as supplementary material instead of being part of the article.

Author Response

Please see attached for detailed comments

This manuscript is a resubmission of an earlier submission. The following is a list of the peer review reports and author responses from that submission.